# Changes in soil phosphorus dynamics amended with rock phosphate-enriched compost and chemical fertilizers

Dilkhush Meena[1], Murli Dhar Meena ⬤[2]*, Satendra Singh Sengar[1], Vidhya Nand Mishra[1], Anup Kumar[1], Mohan Lal Lakhera[1], Mohan Lal Dotaniya[2], Mukesh Kumar Meena[2], Sujith Kumar[2], Bheeru Lal Meena[2], Ram Swaroop Jat[2], Lalit Krishan Meena[2], Mukesh Prajapat[1], Hari Singh Meena[2], Avijit Ghosh[3], Vasudev Meena[2], Pramod Kumar Rai[2], Parvender Sheoran[4], Vijay Singh Meena[5]

**1** Indira Gandhi Krishi Vishwavidyalaya, Raipur, Chhattisgrah, India, **2** ICAR - Directorate of Rapeseed-Mustard Research, Sewar, Bharatpur, India, **3** ICAR-Indian Grassland and Fodder Research Institute, Jhansi, Uttar Pradesh, India, **4** ICAR-Agricultural Technology Application Research Institute, Ludhiana, Punjab, India, **5** ICAR-Indian Agricultural Research Institute, New Delhi, India

\* murliiari@gmail.com, MD.Meena@icar.gov.in

## Abstract

To adopt effective phosphorus (P) management strategies for sustainable mustard (*Brassica juncea L.*) production, it is crucial to understand how P is transformed and made available in the soil by utilising diverse P sources. In this study, a field experiment carried out with five treatment combinations; T1: control, T2:100% recommended dose of fertilizers (RDF), T3: enriched compost @ 6 t/ha, T4:100% RDF + enriched compost @ 6 t/ha, and T5:50% RDF + enriched compost @6 t/ha. Findings indicated that the soil treated with 100% RDF together with enhanced compost at 6 t/ha exhibited substantial improvements in saloid-P, aluminum-P (Al-P), iron-P (Fe-P) and calcium-P (Ca-P) in contrast to compost and 100% RDF alone; however, it was statistically similar to a treatment obtaining 50% RDF + enhanced compost @ 6 t/ha. For soil that has been fertilized with chemical fertilizers and enriched compost, remarkable improvement (15.5% and 8%) in alkaline phospha-tase activity (ALPA) was seen in contrast to the exclusive use of chemical fertilizers and enriched compost, respectively, compared to 100% RDF, enriched compost and chemical fertilizers enhanced the amount of soil microbial biomass phospho-rous (MBP; 55%), microbial biomass carbon (MBC; 34%), and dehydrogenase activity (DHA; 45%). Enriched compost @ 6 t/ha maintained greater P availability and microbial activities culminating in significantly higher mustard grain yield (2.93 Mg/ha). Mustard grain yield was 16.7% higher on soil treated with 100% RDF than control. The outcomes demonstrated that enrichment of processed compost (6 t/ha) with chemical fertilizers (100% RDF) is a good strategy for improved P accumulation deciphering P conversions in soil-plant systems and sustaining mustard yields in degraded ecosystems.

**Data availability statement:** All relevant data are within the paper.

**Funding:** The author(s) received no specific funding for this work.

**Competing interests:** The authors have not declared any conflict of interest.

**Abbreviations:** RDF, recommended dose of fertilizers; Al-P, aluminium-P; Fe-P, iron-P; Ca-P, calcium-P, ALPA, alkaline phosphatase activity; MBP, microbial biomass phosphorous; DHA, dehydrogenase activity; MBC, microbial biomass carbon, total-P, total phosphorus; inorganic-P, inorganic phosphorus; org-P, organic phosphorus; FYM, farmyard manure; SOC, Soil organic carbon; RP, Rock phosphate; Mt, million tonnes

## Introduction

Lower and inefficient fertilizer use, precipitation, fixation, and leaching losses collectively contribute to the highly variable and limited availability of phosphorus (P) in soil [1]. Since inorganic phosphorus (Pi) in the soil is the main source of phosphorus (P) for plant uptake, it is essential to comprehend the different fractions of phosphorus to understand its availability in the soil-plant system. To enhance plant growth, it is vital to comprehend soil P availability and the interconversion of Pi fractions from different P pools [2]. Inorganic P comprises calcium-bound P (Ca-P), iron-bound P (Fe-P), aluminium-bound P (Al-P), and P within the matrices of P-retaining of occluded P. In Indian soils, Pi constitutes 54–84% of the total P content [3]. According to [4], Olsen-P is a highly sensitive indicator for determining the amount of soil P available for plant growth. Organic P (Org-P), found mainly in organic matter, must undergo mineralization to release inorganic P before plant absorption. This mineralization process, which releases P into the soil solution, is influenced by soil microbial activity and the production of enzymes associated with Org-P hydrolysis [5]. Phosphatase and phytase hydrolyze the ester bonds between carbon and phosphorus (C-O-P) during the mineralization of organic P [6,7] and [1]. Phosphatase activity is crucial in determining the rate of soil organic P fraction mineralization and serves as a good predictor of P deficiency [8]. Explored the hydrolysis of a range of low-molecular-weight P compounds by alkaline phosphomonoesterases [9]. These results suggest that the P cycle amongst soil and plant nutrition depends on enzymes [10]. For crop production, the usage of chemical fertilizers has increased substantially all over the world, but their availability to first crop decreased over time due to their high-water solubility, which in turn increases the possibility of precipitation, fixation, adsorption, and leaching losses [1]. Total fertilizer nutrient use in India has increased from a meagre 69,800 tonnes in 1950–1951 to 28.28 million tonnes (Mt) in 2010–2011 [11]. Nitrogen (N), P, and potassium (K) minerals are being consumed in amounts of 16.89, 8.0, and 3.39 Mt, respectively. India uses a lot of fertilizers in its agriculture, without which it would not be able to produce the food grains and other agricultural commodities needed to sustain its ever-growing population of India.

The second most crucial macronutrient for crop development and yield is phosphorus (P). However, in India, phosphatic fertilizers are very costly. This is mostly because of the cost of importing raw ingredients, like sulphur and high-quality rock phosphate (>30% $P_2O_5$), essential for producing phosphatic fertilizers for commercial use. Rock phosphate (RP) production in India is about 160 Mt [11]. Unfortunately, most of them fall into the category of low-grade materials since they contain less than 20% $P_2O_5$, which are deemed unfit for use in the production of phosphatic fertilizers for commercial use. Although these materials perform well in acidic soil, rather in neutral to alkaline soil is very low [6].

Through composting, efforts have been made to use this waste mica and low-quality RP as a significant source of P and K for plants [12]. The utilization of phosphate in rock phosphate by composting technology is crucial because it increases the compost's nutrient content by causing inaccessible phosphorus to change into an available form that plants can utilize [7,13].

Enriched compost generated from farm waste, city trash, and agro-based industrial wastes could be used in place of farmyard manure (FYM) or conventional compost to safeguard the bio-physical quality of the soil. Traditional compost and FYM are deficient in important nutrients, especially P content, which is insufficient to replace chemical fertilizers [14]. Compost that has been enhanced helps in cutting down on transportation costs.

For a longer period, more P is retained in the soil solution with RP-enriched compost than water-soluble P-fertilizer. But in addition to being a low-input technology, using immature compost in agriculture results in fewer issues including immobilisation, less seed germination inhibition, reduced plant growth, and crop damage brought on by phytotoxic chemicals during composting [15–17]. Composting is a sustainable method of turning waste into wealth that can be used as an organic fertilizer for agriculture. The biological process of composting turns biodegradable organic materials into a material that resembles humus by using naturally occurring bacteria. It is believed that applying organic manures directly to the soil will boost the availability of P from RP. Organic compounds break down into many different organic acids, especially the hydroxyl ones (byproducts of microbial metabolism). These acids release P by dissolving insoluble metal phosphates and chelating metal ions [1,6].

Mustard (*Brassica juncea*) is a key oilseed crop in India, playing a dominant role in the agricultural economy. It ranks first in oilseed production and holds a prominent position in the edible oil sector of India [18]. In order to produce mustard (*Brassica juncea L.*) sustainably, an effort was undertaken to monitor the effective use of mustard stover and local sources of phosphorus. Mustard stover, Udaipur rock phosphate (RP), mica, gypsum, and Aspergillus awamori were used to create RP-enriched compost. Through an experiment conducted in the field with mustard cultivation, these materials were assessed. Therefore, it is imperative to how various fertilizer management techniques affect P cycling of soil. This effort is primarily focused on developing a unique organic soil amendment that might potentially replace expensive chemical P fertilizers. The findings mentioned before suggested that maintaining the ideal P levels in soil requires the integration of compost and chemical fertilizers. However, there is not enough data available about the P fractions in soils that have been modified with various sources. Therefore, the purpose of the current study is to explore the effects of chemical fertilizers and enriched compost on different pools of P, (ii) examine mustard crop yields, and (iii) evaluate changes in soil biological properties. It was our hypothesis that applying enriched compost would raise the P fractions and crop production of mustard.

## Materials and methods

### Research site and weather

The field experiment took place at the ICAR-Directorate of Rapeseed-Mustard Research farm in Sewar, Bharatpur, Rajasthan, India. This farm is geographically located at 27.150 N latitude and 77.300 E longitude, with an elevation of approximately 178.37 meters above mean sea level (MSL). During the cropping season from September 2021 to March 2022, the average rainfall was recorded at 288 mm. The region experiences a semi-arid and subtropical climate, characterized by very cold winters and hot, dry summers. The soil at the site has a sandy loam texture. The initial physico-chemical properties of the experimental site are detailed in Table 1.

### Compost preparation

In order to make enriched compost for the field experiment, waste mica, gypsum, Udaipur rock phosphate, mustard stover, and *Aspergillus awamori* were used for every 100 kg of rice straw, 50 g of fresh *Aspergillus awamori* mycelia and the appropriate proportions of waste nutrients were added to the mustard stover [6]. After 120 days of composting, the final compost's chemical composition is shown in Table 2.

### Chemical characterization of composts

After a composting period of 120 days, mature fresh compost samples were collected for analysis. The chemical properties analyzed included total carbon (C), nitrogen (N), phosphorus (P), potassium (K), and sulfur (S). Total C content was determined using the ignition method described by [19].

**Table 1. Initial soil characteristics (physical, chemical, and biological) in the experimental farm.**

| Parameters | Values |
|---|---|
| pH (1:2.5) | 8.3 |
| EC (dS/m) | 2.2 |
| Ammonical- N ($NH_4^+$-N) (kg/ha) | 15.4 |
| Nitrate-N ($NO_3^-$-N) (kg/ha) | 27.2 |
| Available-P (0.5$M$ $NaHCO_3$, pH 8.5) (kg/ha) | 17.5 |
| Available-K (1$N$ $NH_4OAc$) (kg/ha) | 210 |
| Organic C (%) | 0.42 |
| MBC (mg/kg) | 127 |
| DHA (µg TPF/g/h) | 9.78 |
| AKP (µg PNP/g/h) | 122 |

**Table 2. Chemical characterization of enriched compost.**

| Parameters | Enriched compost |
|---|---|
| Total P (%) | 3.5±0.01 |
| Total K (%) | 2.7±0.11 |
| Total S (%) | 0.55±0.03 |
| Total carbon (%) | 25.7±0.29 |
| Total N (%) | 1.0±0.07 |
| C/N | 25.7±0.02 |
| $pH_w$ (1:5) | 7.8±0.10 |

## Details of the field trial and treatment

The impact of chemical fertilizers and enriched compost in changes of P fraction and how they affect mustard yield. There were five treatments applied: T1: No enriched compost and fertilizer application (control); T2: 100% of the N:P:K:S recommended dosage of chemical fertilizers (RDF) at 80:40:40:40 kg/ha; T3: 6 t/ha of enriched compost; T4: 6 t/ha of 100% RDF plus enriched compost; T5: 6 t/ha of 50% RDF plus enriched compost. The study was set up using a randomized block design (RBD) with four replications, each maintaining a 6 x 4.5 m² plot. Giriraj (DRMRIJ-31) was a mustard variety that was sown in October and harvested in March. The prescribed package of practices was followed for all other agronomic practices, such as weed control and plant protection measures.

## Soil analysis

Soil samples were collected from the top 0–15 cm layer after harvesting the mustard crop. To test microbiological activity, 100 g of moist soil from each plot was immediately stored in a refrigerator at 4 °C. Post-harvest, soil samples were air-dried, ground with a wooden pestle, and sieved through a 2-mm mesh. Various inorganic P fractions were determined using a modified P fractionation scheme by [20], Olsen-P [21], and soil organic carbon [22]. Total P, inorganic P, and organic P (Po) were determined following the [23] method. To estimate total P (Pt), a 1.0 g sample was weighed into a silica crucible and burned at 550 °C for one hour in a muffle furnace. Organic P (Po) was calculated using the formula: inorganic P in unignited soil minus total P in ignited soil equals organic P (Po).

Dehydrogenase activity (DHA) was evaluated using the method by [24], while alkaline phosphatase activity was measured as described by [25]. Microbial biomass carbon (MBC) was measured using the fumigation-extraction method [26]. Microbial biomass P was calculated using the formula provided by [27].

## Statistical analysis

Duncan's multiple-range tests were used to analyze data from field studies that had a probability of less than 0.05 [28]. Using the SPSS program, the least significant difference (LSD) between means was determined (SPSS version 16.0). A structural equation model was developed to understand the pathways of influence of enriched compost on mustard yield with four latent variables. The model's appropriateness was evaluated using the $\chi2$-test, the root mean squared error of approximation (RMSEA) index, and the goodness of fit (GFI) index. To determine if the model could reasonably explain the data's patterns, the $\chi2$ statistic was employed. Favourable model fits were revealed no significant difference on the $\chi2$-test ($P < 0.05$), high GFI ($> 0.9$), and low RMSEA ($< 0.08$). The absolute effects of latent variables dictated the width of the arrows.

## Results

### Changes in Olsen-P

Treatments utilizing compost materials in conjunction with permitted chemical fertilizer dosages either alone or in combination resulted in significantly ($p < 0.05$) higher Olsen-P. The concentration of Olsen-P ranged from 16.46 to 33.33 kg/ha after the mustard harvest (Fig 1). Following the harvest of mustard, the application of 100% RDF resulted in a considerable rise in Olsen-P compared to control. Olsen-P was considerably elevated by the combined use of 50% RDF and enriched compost as opposed to 100% RDF alone.

Olsen-P concentration was reported to be significantly higher in the treatment receiving 100% RDF with enriched compost at 6 t/ha; this treatment's concentration was 101.9% higher than the unfertilized plot (control). Notably, the highest possible concentration of Olsen-P was achieved when 100% RDF was integrated with enriched compost. Treatment receiving 100% RDF resulted in about 62% higher Olsen-P than unamended soil.

### Total P, Inorg-P and Org-P

After the mustard crop was harvested, the integration of chemical fertilizers and enriched compost improved the total P concentration in the soil considerably ($p < 0.05$) when in comparison to control. The highest amount of total P (612 mg/kg) had reported with 100% RDF along with enriched compost ($T_4$) after mustard harvest (Fig 2) and it was significantly higher

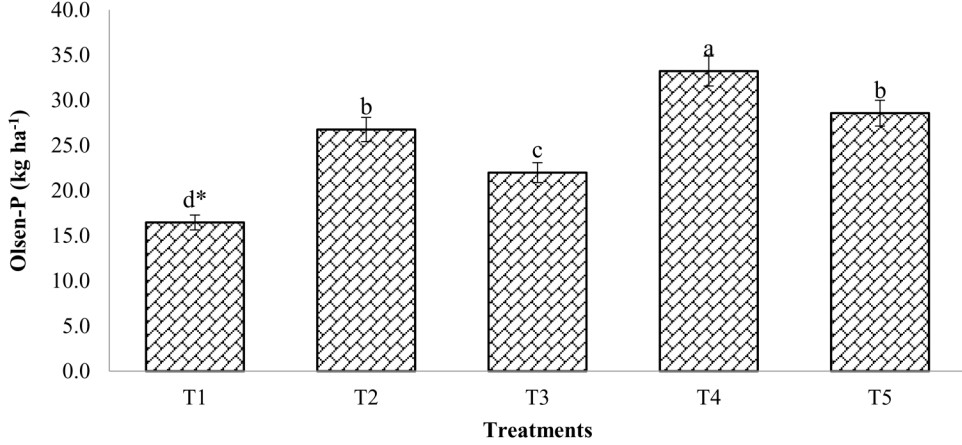

**Fig 1. Changes in Olsen-P as amended with enriched compost and chemical fertilizers.** ($T_1$: control; $T_2$: 100% RDF; $T_3$: enriched compost; $T_4$: 100% RDF + enriched compost; $T_5$: 50% RDF + enriched compost). *** The least significant difference test for separation of means indicates that means that are separated by the same letter are not significantly distinct at P=0.05.**

over alone use of 100% RDF and control plots. There was a 14.4% increase in total P over control in soil that received 100% RDF (T2). Treatment T5, on the other hand, produced a total P that was 21.5% more than control.

With the exception of control, there were no discernible changes between the treatments in terms of inorg-P. Compared to integrated compost use at 6 t/ha + 100% RDF, there was a little decrease in inorg-P recorded when using solely enriched compost (Fig 2). When soil treated with 100% RDF and enriched compost at 6 t/ha was compared to chemical fertilizers alone, there was a significant build-up in inorganic P, which was 34.6% greater than the control. It was clear that applying 100% RDF (T2) increased inorganic-P by 19% compared to the control.

Application of enriched compost, either alone or in conjunction with chemical fertilizers, more organic P was obtained even though there was no appreciable difference in organic P between the treatments. (Fig 2). Compared to the control, the organic-P rose by 19.5% in T4 and 7.8% in T2. The treatments showed statistical similarities with respect to organic-P; nevertheless, the use of enriched compost outperformed the use of 100% RDF alone.

## P fractionations in sequence (S-P, Fe-P, Al-P and Ca-P)

After the mustard harvest, field research clearly shows that there was a significant (P < 0.05) increase in organic P percentages in all fertilized plots compared to the unfertilized plot (T1) (Table 3). Higher amounts of saloid-P (S -P) were

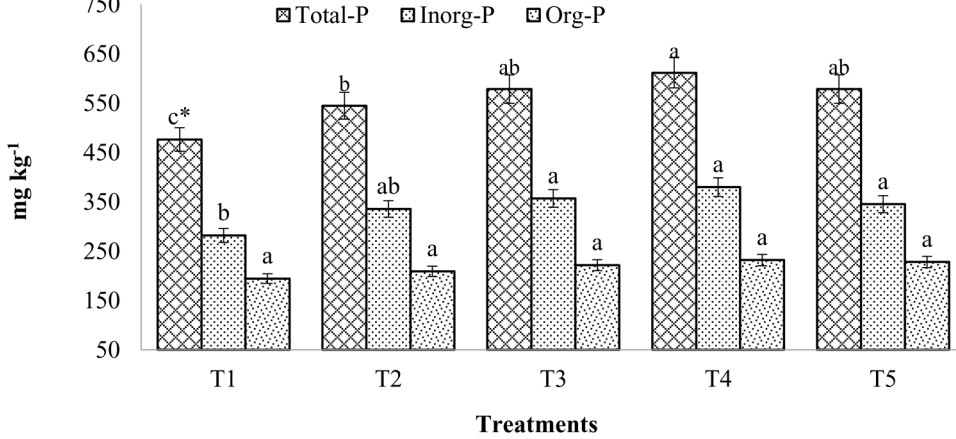

**Fig 2. Changes in total-P, inorganic-P and organic-P fractions as affected with compost and chemical fertilizers.** (T1: control; T2: 100% RDF; T3: enriched compost; T4: 100% RDF + enriched compost; T5: 50% RDF + enriched compost). ***The least significant difference test for separation of means indicates that means that are separated by the same letter are not significantly distinct at P = 0.05.***

**Table 3. Changes in inorganic P fractions as amended with compost and chemical fertilizers.**

| Treatment | Saloid- P mg/kg | Fe-P mg/kg | Al-P mg/kg | Ca-P mg/kg |
|---|---|---|---|---|
| T1 | 10.1[e]* | 10.3[c] | 14.2[c] | 168[c] |
| T2 | 20.5[c] | 20.7[b] | 22.3[bc] | 210[b] |
| T3 | 18.2[d] | 20.4[b] | 20.3[bc] | 212[ab] |
| T4 | 25.3[b] | 25.1[a] | 31.4[a] | 233[a] |
| T5 | 23.3[a] | 22.3[ab] | 23.4[ab] | 219[ab] |
| LSD (P = 0.05) | 1.9 | 3.9 | 9.7 | 24.7 |

T1: Control; T2: 100% RDF; T3: Enriched compost; T4: 100% RDF + Enriched compost; T5: 50% RDF + Enriched compost.

*The least significant difference test for separation of means indicates that means that are separated by the same letter are not significantly distinct at P = 0.05.

found in the soil of treatments that received enriched compost and chemical fertilizers. The integrated use of compost and chemical fertilizers produced significantly larger levels of S-P (25 mg/kg) compared to 100% RDF and control. Table 3 shows that as compared to the control treatment, a significantly greater iron bound-P (Fe-P) had been reported with 100% RDF. S-P varied from 10 to 25 mg/kg however, with the exception of T5, the application of enriched compost at a rate of 6 t/ha combined with 100% RDF had maintained a much higher S-P (25.1 mg/kg).

When applied either independently or in conjunction with RDF, enriched compost produced noticeably higher Aluminum bound-P (Al-P) than the control (Table 3). Similarly, Al-P was considerably greater in 100% RDF than in control. In contrast to other treatments, enriched compost at a rate of 6 t/ha combined with 100% RDF had a much higher Al-P level. Although it was statistically similar to enriched compost at 6 t/ha plus 50% RDF, the adoption of enriched compost at 6 t/ha plus 100% RDF produced significantly higher levels of Al-P than other treatments. There was a notable ($p < 0.05$) increase in calcium bound-P (C-P) across treatments that received chemical fertilizers and enriched compost compared to the control (Table 3). Of the various P fractions in soil, Ca-bound P dominated regardless of treatments. Ca-P levels in 100% RDF were 20% higher than in the control.

### Microbial biomass phosphorus (MBP) and Microbial biomass carbon (MBC)

Following mustard crop harvest, MBP was significantly increased due to the integration of chemical fertilizers and enriched compost (Fig 3). Significantly higher MBP was seen with 100% RDF with enriched compost at 6 t/ha compared to the control. The maximum MBP (5.83 mg/kg) in soil was found when 100% RDF and compost were used together. This was 55% higher than when 100% RDF was used alone.

Fig 4 provides information on microbial biomass carbon (MBC). Substantial differences ($p < 0.05$) existed between the treatments. MBC increased substantially by 23.98% in the 100% RDF treatment compared to the control. The results demonstrated a considerable increase in MBC compared to unfertilized plots when compost was treated, either alone or in conjunction with chemical fertilizers (T4 and T5). The highest amount of MBC was observed with T4 (206 mg/kg).

### ALPA and DHA

Fig 5 shows the alkaline phosphatase activity (ALPA). When chemical fertilizers and compost were added to the soil, the ALPA levels increased substantially compared to the control. However, following the harvest of the mustard crop, a

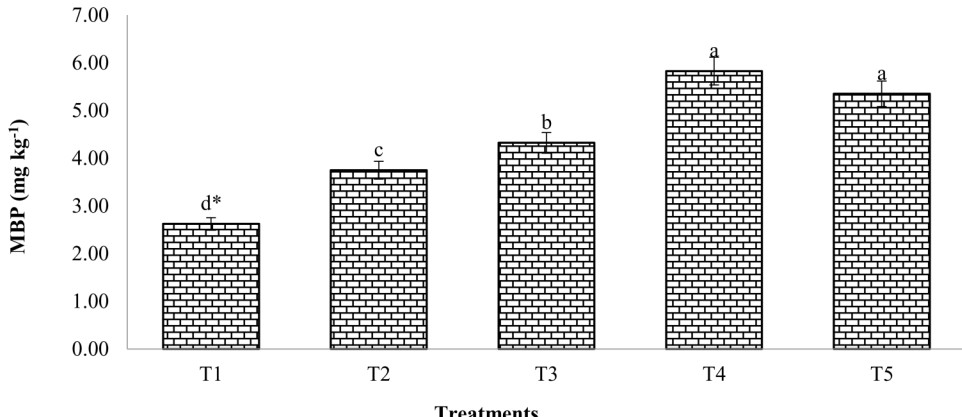

**Fig 3. Effect of compost and chemical fertilizers on microbial biomass phosphorus.** ($T_1$: control; $T_2$: 100% RDF; $T_3$: enriched compost; $T_4$: 100% RDF + enriched compost; $T_5$: 50% RDF + enriched compost). *** The least significant difference test for separation of means indicates that means that are separated by the same letter are not significantly distinct at P = 0.05.**

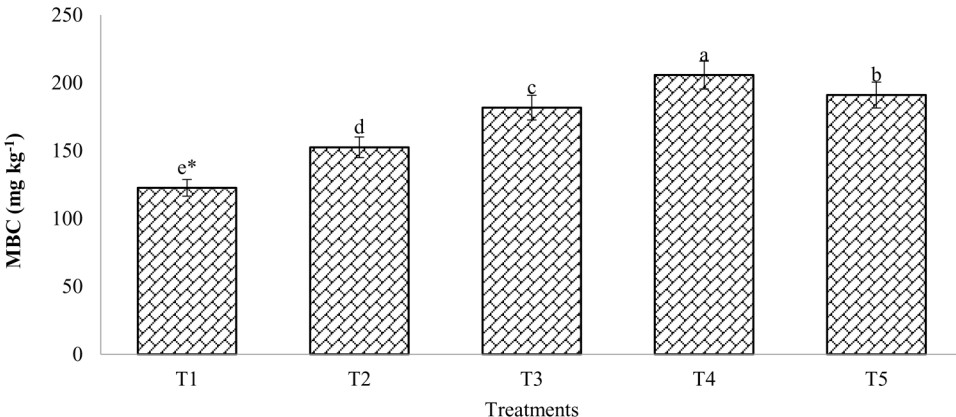

**Fig 4. Changes in microbial biomass carbon (MBC) as affected by compost and chemical fertilizers.** ($T_1$: control; $T_2$: 100% RDF; $T_3$: enriched compost; $T_4$: 100% RDF + enriched compost; $T_5$: 50% RDF + enriched compost). *The least significant difference test for separation of means indicates that means that are separated by the same letter are not significantly distinct at P = 0.05.*

significant distinction was noted between the control and 100% RDF treatments. After the mustard crop was harvested, treatment T4 (159 µg PNP/g soil/h) showed a substantially higher ALPA activity than the other treatments, except T5 (149 µg PNP/g soil/h) and T3 (144 µg PNP/g soil/h) (Fig 5). In comparison to the control, treatment T5 generated 29.1% more ALPA in the soil, while treatment T2 maintained a 19.5% higher ALPA in the soil. In general, ALPA was more influenced with enriched compost than control.

The findings show that, following the mustard harvest, there was no discernible difference in the levels of dehydrogenase activity (DHA) between the control and the 100% RDF application. Results indicated that the integration of chemical fertilizers T4 (100% RDF) and enriched compost (at 6 t/ ha) had higher DHA (18.0 µg TPF/g soil/h) than other treatments

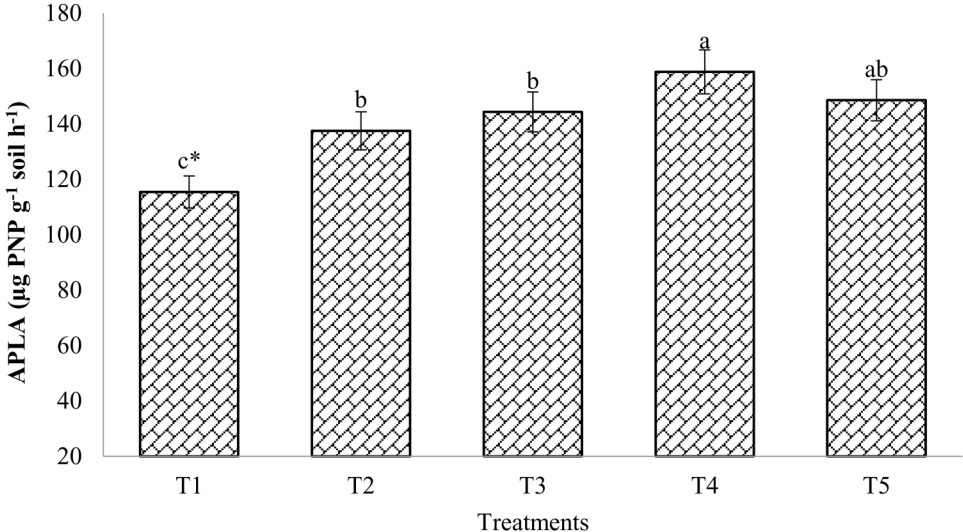

**Fig 5. Effect of enriched compost and chemical fertilizers on alkaline phosphatase activity.** ($T_1$: control; $T_2$: 100% RDF; $T_3$: enriched compost; $T_4$: 100% RDF + enriched compost; $T_5$: 50% RDF + enriched compost). *The least significant difference test for separation of means indicates that means that are separated by the same letter are not significantly distinct at P = 0.05.*

(Fig 6). About a 24% increase in dehydrogenase activity was seen after applying 100% RDF compared to the control, while a 79.7% increase was reported in treatment T4.

## Soil organic carbon (SOC)

The impact of organic treatments on soil organic carbon was higher than that of chemical fertilizer application alone. The findings showed that 100% RDF had less of an impact on SOC than 100% RDF when mixed with enriched compost (Fig 7). However, there were no noted appreciable variations between the treatments.

Treatments getting compost at 6 t/ha coupled with 100% RDF build up more SOC than other treatments. In comparison to 100% RDF alone, the magnitude of changes in soil organic carbon was greater in plots treated with compost. SOC

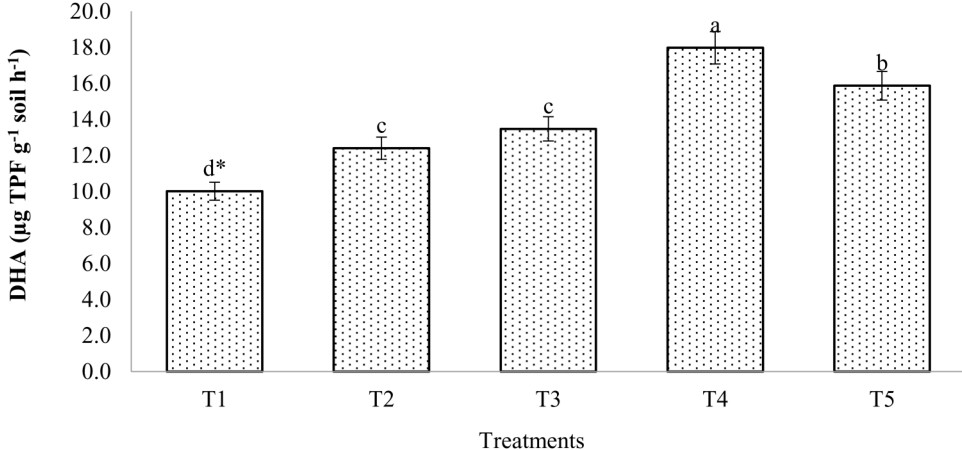

**Fig 6. Effect of enriched compost and chemical fertilizers on dehydrogenase activity.** ($T_1$: control; $T_2$: 100% RDF; $T_3$: enriched compost; $T_4$: 100% RDF + enriched compost; $T_5$: 50% RDF + enriched compost). *The least significant difference test for separation of means indicates that means that are separated by the same letter are not significantly distinct at P = 0.05.*

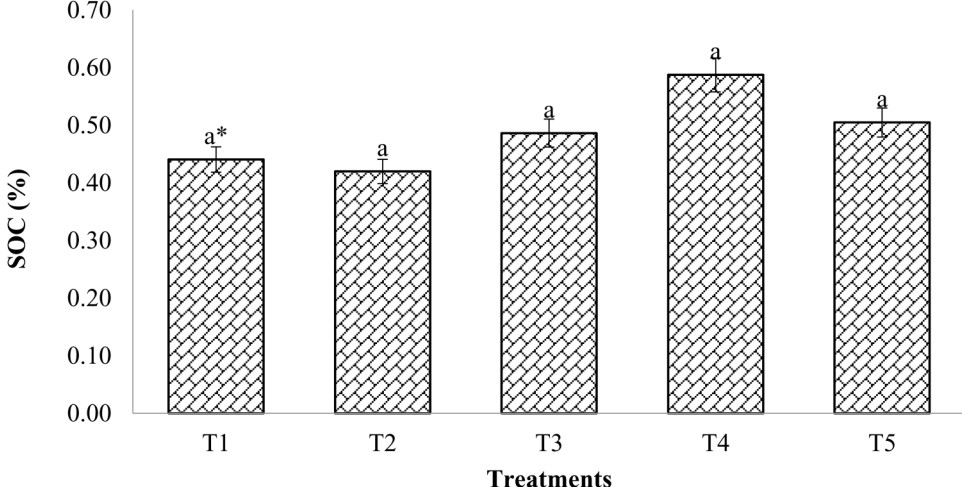

**Fig 7. Changes in soil organic carbon as influenced with compost and chemical fertilizers.** ($T_1$: control; $T_2$: 100% RDF; $T_3$: enriched compost; $T_4$: 100% RDF + enriched compost; $T_5$: 50% RDF + enriched compost). *The least significant difference test for separation of means indicates that means that are separated by the same letter are not significantly distinct at P = 0.05.*

varied from 0.44–0.59% under various treatments. The highest SOC (0.59%) was found under the treatment receiving 100% RDF with enriched compost at 6 t/ha (T4).

### Grain and straw yield

Mustard yield was considerably (p<0.05) influenced with enriched compost and chemical fertilizers (Fig 8). When chemical fertilizers were combined with enhanced compost, mustard's grain and stover yields improved noticeably. Additionally, a considerably greater grain (2.93 Mg/ha) yield of mustard was found upon integration of 100% RDF plus enriched compost at 6 t/ha in comparison to the unfertilized plot. When compared to control, the suggested fertilizer dose (T2) increased grain and stover yield by 41% and 66%, respectively.

The grain (2.84 Mg/ha) and stover (4.70 Mg/ha) yield of the mustard crop was higher under treatment (T5) receiving enriched compost at 6 t/ha along with chemical fertilizers (50% RDF) compared to T2 and control. Treatment T5 recorded 57 and 74% higher grain and stover yield of mustard, respectively over control. The total (Grain+Stover) yield of mustard ranged from 6.13–7.86 Mg/ha during the study period. All treatments varied from 11 to 28% higher total mustard yield. Higher grain, stover, and total yield were obtained by using enriched compost in combination with chemical fertilizers as opposed to compost.

## Discussion

### Changes in Olsen-P

Mustard crops were more capable of obtaining P whether it was applied to either chemical fertilizers (RDF) or organic amendments (enriched compost). Compared to other treatments, treatments receiving 100% RDF in addition to enriched compost at 6 t/ha showed a significant build-up in Olsen-P. It can be explained that the addition of organic materials, such as manure also provided significant amounts of P to the soil, resulting in a higher build-up of Olsen-P in soil as treated with enriched composts than control [1,7]. When compared to the usage of compost and chemical fertilizers alone, Olsen-P significantly enhanced with the integration of organic and inorganic fertilizers. Similar findings indicated that P (Olsen-P) availability improved over time in the ensuing years and was related to compost mineralization. Similar

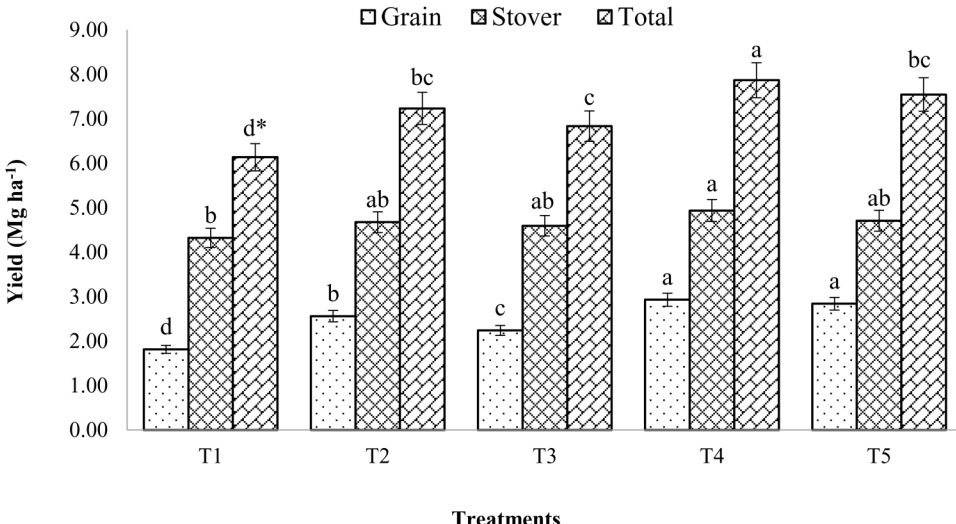

**Fig 8. Effect of enriched compost and chemical fertilizers on mustard yield.** (T$_1$: control; T$_2$: 100% RDF; T$_3$: enriched compost; T$_4$: 100% RDF+enriched compost; T$_5$: 50% RDF+enriched compost). *The least significant difference test for separation of means indicates that means that are separated by the same letter are not significantly distinct at P=0.05.*

increases in soil fertility, including Olsen-P, have been attributed to the long-term use of composts [29]. Additionally [30], reported that releasing of humic acid, the breakdown of organic matter supplemented with 1% of compost treatment could transform unavailable soil phosphates into available form (Olsen-P). According to the findings, utilizing enriched compost in conjunction with mineral fertilizers increased P mineralization, making it available to plants for an extended period than chemical fertilizers [31].

## Total P, inorganic P and organic P

Applying enriched compost at 6 t/ha, either alone or combined with 100% recommended dose of fertilizers (RDF), significantly enhanced total P and inorganic P compared to using 100% RDF alone. This is attributed to the more consistent supply of P from enriched compost compared to chemical fertilizers alone [6,7]. Higher total P in soil reported with composted sewage sludge and decomposing rice straw compared to untreated soils. In the control group, which received no additions, P levels were lower due to crop depletion compared to the 100% RDF treatment [32]. Compost-rich in rock phosphate increased alkaline and acid phosphatase activity, as well as total, inorganic, and water-soluble P levels [31]. More inorganic P was found in soils treated with enriched compost [6] compared to the control. The highest inorganic P concentration was observed in soils treated with 100% RDF and enriched compost at 6 t/ha. These results suggest that integrating organic and inorganic P sources could be more beneficial for sustainable crop production [1].

Bacterial alkaline phosphatase, fungal acid phosphatase, and root-borne acid phosphatase enzymes play a key role in decomposing organic P in the rhizosphere. Organic P is a significant source of P for plants [33]. The addition of enriched compost led to greater organic P accumulation than the use of chemical fertilizers alone, indicating that enriched compost supplies more organic P. Although the amounts of organic P in each treatment were similar, treatments with enriched compost, either alone or combined with chemical fertilizers, had higher levels of organic P. This could be due to the gradual mineralization of the organic pool over time from the enriched compost. These findings are consistent with other studies [29,31].

## Sequential fractionations of P (S-P, Fe-P, Al-P and Ca-P)

The best technique for examining soil inorganic P interconversions across various P pools is inorganic P fractionation [34]. P availability is influenced by the pH of the soil because it determines the amount and presence of metal cations (Ca, Fe, and Al) that are likely to precipitate with P ions in the soil solution. Most P ions precipitate as dicalcium or octacalcium phosphates in neutral to alkaline soils; hydroxyl apatite and least soluble apatites follow [4,35]. According to this study, enhanced compost and chemical fertilizers kept soil concentrations of all P fractions, Saloid-P, Al-P, Ca-P, and Fe-P higher than control plots [7]. However, a number of variables influence the availability of P in soils, such as the presence and amount of metal cations that are prone to precipitate with P ions in the soil solution (Fig 9).

However, saloid P (S-P) has been suggested to be the most crucial P fraction for plant growth [36]. The findings showed that all P percentages increased relative to the control when compost and chemical fertilizers were added to the soil. When enriched compost and chemical fertilizers were combined, the level of all inorganic P components was considerably higher than it was in the control. On the other hand, soil treated with chemical fertilizers, increased compost, and their integration showed a rise in inorganic P fractions when compared to control. Regular applications of animal dung and chemical fertilizers increased the levels of both labile and non-labile P pools in soils in natural agroecosystems [37]. The findings revealed that combining phosphorus from organic and inorganic sources improved phosphorus solubility, reduced phosphorus precipitation in the soil, and increased phosphorus availability for mustard crops.

## Microbial biomass P (MBP) and Microbial biomass carbon (MBC)

According to field studies, adding enriched compost to soil treated with compost and mineral fertilizers caused in a considerable surge in microbial biomass P (MBP) as compared to using simply organic and inorganic sources of P [38,39]. MBP

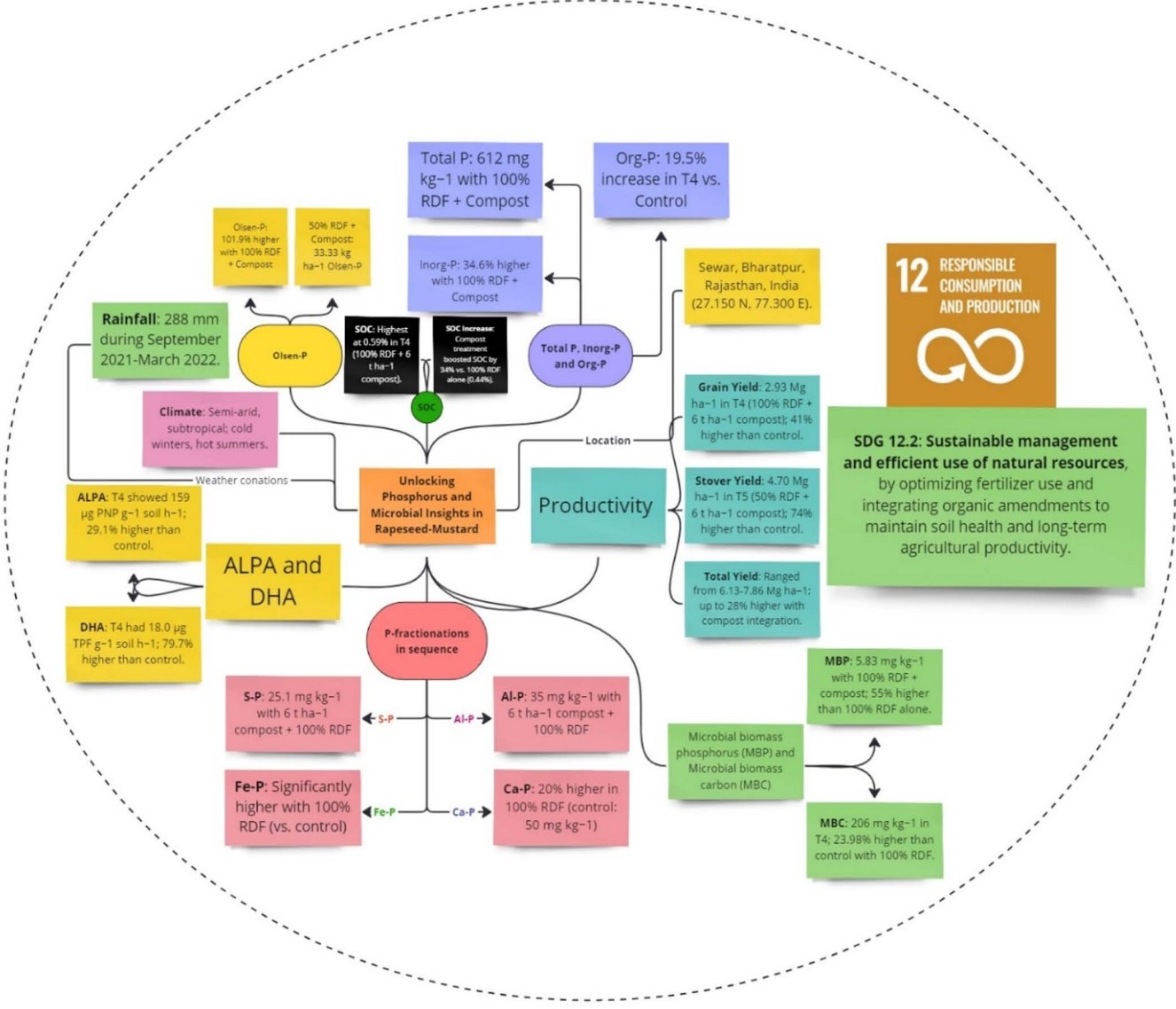

**Fig 9. Effect of sustainable management of natural resources on different pools of P and soil health.**

is considered to be among the key indicators of soil phosphorus availability, particularly for P-deficient soils. MBP is a vital source of labile P in soil and is essential for the production of P trans between organic and inorganic components. MBP produces 2% to 10% of the total phosphorus in soil, which is greater than Olsen-P and the annual P flux from the turnover of microbial biomass, according to [40]. Microbial biomass carbon (MBC), a crucial component of soil organic carbon, is more impacted by management techniques than changes in the overall quantity of soil organic matter [14,41–43]. Significant ($p<0.05$) MBC levels were observed in the treatment receiving enriched compost compared to the control. However, much more MBC was observed in compost and chemical fertilizers were used together compared to other treatments, this can be clarified by the fact that combining organic and inorganic materials produced more MBC than using mineral fertilizers alone [6,44].

## Alkaline phosphatase activity (ALPA) and dehydrogenase activity (DHA)

Phosphatases are stimulated enzymes that soil organisms and plant roots release; they can be activated by P deprivation [45]. Phosphatase activity is essential to sustain and regulate the rate of soil organic P mineralization [8]. The retention of acid phosphatase activity in unmodified control soil supported the result that mineral fertilizer reduced this affinity or altered the composition and activity of soil microbiota, whereas organic manure/compost did not affect enzyme-substrate affinity [46].

A group of endocellular enzymes known as dehydrogenase activity (DHA) is accountable for catalyzing the oxidation of soil organic matter. It is a crucial indicator of the total amount of microbial activity present in soil [47]. It occurs intracellularly in all living cells and is associated with microbial respiratory processes [14,48,49]. The current findings imply that ongoing use of enriched compost affects soil enzyme activity, which in turn controls how nutrients are changed in soil, either directly or indirectly. In comparison to control, DHA significantly improved in treatments that included compost and mineral fertilizers [50].

## Soil Organic carbon (SOC)

It is often known how important organic matter is to the soil and how it regulates numerous environmental parameters. When 100% RDF and compost were combined at a rate of 6 t/ha, the SOC content increased relative to the control; this was clearly because the compost assimilated more biomass. [14,51]. No significant variations in SOC between the treatments were found in this investigation. This could be the result of the organic substance being first disintegrated into smaller components so that plants and soil microorganisms can use it [52]. Although the microbial population and rate of compost decomposition to SOC are governed by its chemical nature [53], our findings demonstrated that the combination of RDF and chemical fertilizers produced superior in soil than other treatments. SOC had considerably raised in enriched compost treated soil compared to control, whether it was applied alone or along with chemical fertilizers [54].

## Seed and straw yield

The study found that enriched composts and mineral fertilizers significantly affect mustard production (p < 0.05). Mustard yields were higher than those of the control, which can be attributed to the application of enriched compost and the recommended dosage of chemical fertilizers. Higher crop yields observed with the integration of compost and chemical fertilizers could be attributed to increased microbial activity, better supply of secondary and micronutrients, which are not supplied by 100% RDF alone, and lower nutrient losses from the soil, among other benefits of organic matter beyond P and K supply [29,55]. The improved mustard production may have also been influenced by the better soil physical properties in plots treated with enriched compost. Others have found similar outcomes in enhancing soil physical properties as a result of adding organic amendments [56,57].

The SEM analysis revealed that compost properties equally impacted soil microbial properties and P transformation. However, microbial factors contributed significantly to changing the soil P fractions (Fig 10). Higher microbial activity resulted in a greater release of organic acids, causing higher desorption of inorganic P. Crop yield was mostly influenced by P transformation. Organic acids released phosphate from iron, aluminium and calcium phosphate compounds when they formed complexes with ionic species of Fe, Al or Ca [58].

This is braced by the rise in plant-available P in soil treated with compost. It's also possible that these methods reduced P hysteresis, which allowed adsorbed P to escape. The phosphate status of the soil is correlated with the amount of water-soluble silicate present in it. The quantity of phosphate that the crop could absorb from the soil rose when the amount of water-soluble silicate in the soil increased [59]. It is well known that silicate can be absorbed by soils from solutions in certain circumstances; the surface of the hydroxylated ferric and aluminium films is most likely the main adsorption site. The production of mustard is significantly impacted by the availability of P.

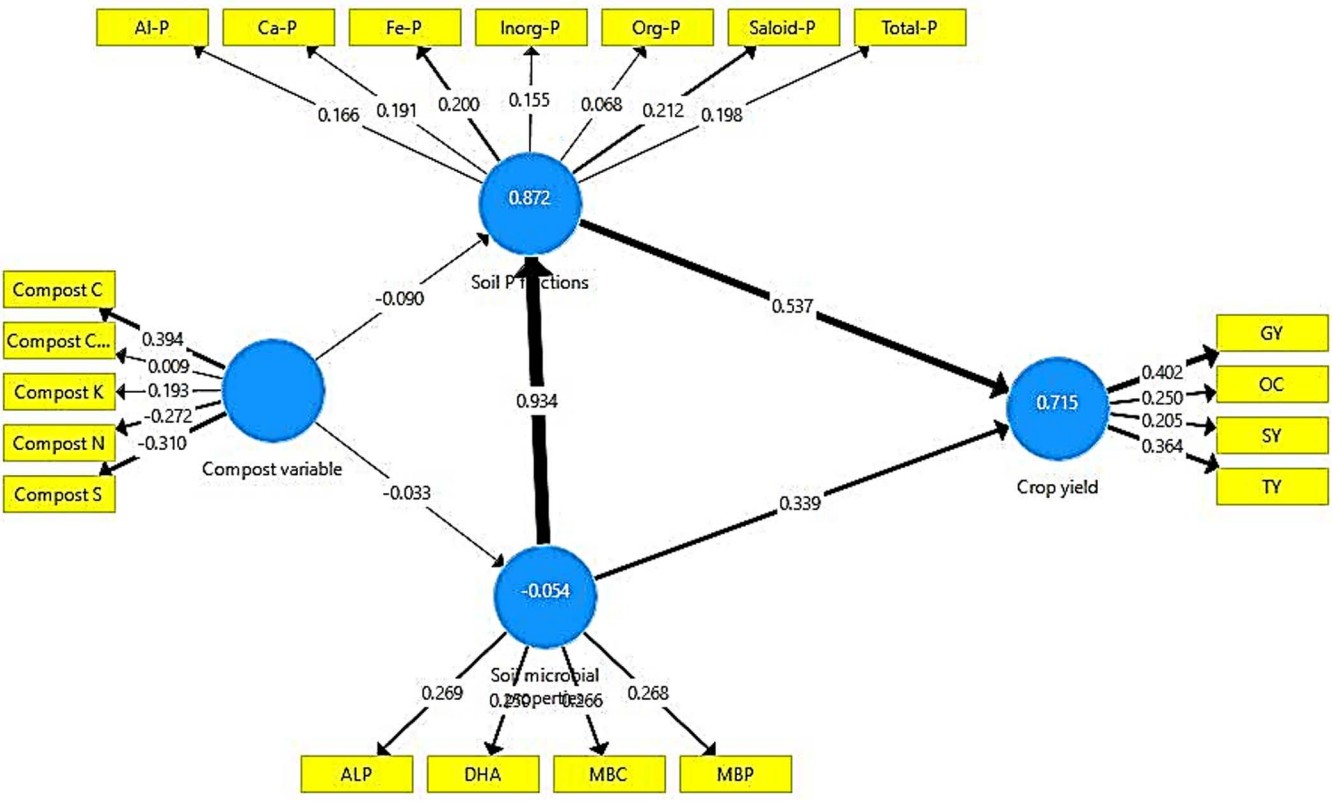

**Fig 10. Path analysis to depict the impact of compost properties, P transformation, and microbial properties on mustard yield. Models satisfactorily fitted to data based on χ2 and RMSEA analyses [χ2 = 96.21, GFI = 0.79, RMSEA < 0.001]. Solid arrows represent the significant effects. The arrow widths indicate the strength of the casual relationship.**

## Policy recommendation for sustainable mustard production

Sustainable management of natural resources is a prime need across the globe. In this connection, a holistic approach will consist of integrated nutrient management, more subsidy on compost and flexibility on policy norms related to compost prepared from waste. However, soil organic matter is a vital concern to improve crop yield and soil health parameters for sustainable agricultural production to feed the growing population. A field experiment was conducted using the organic (enriched compost) and inorganic (chemical fertilizers) to see the effect on mustard yield and soil fertility parameters. Mustard crop yield was highest enhanced up to 2.6 Mg/ha in the treatment (100% RDF + 6 t/ha); however, an overall 11–28% increment was reported (Fig 11).

Soil organic carbon was significantly improved with enriched compost and chemical fertilizers. Soil fertility parameters and biological indicators were reported maximum in the treatment (100% RDF + 6 t/ha). This experiment opens the door to think about subsidizing compost use schemes at the national level to improve the SOC and sustainable crop production potential, particularly in South -East Asian countries.

## Conclusions

The present study demonstrated that addition of enriched compost and chemical fertilizers remarkably improved Olsen-P, total P, and inorganic P as compared to control. Enriched compost in conjunction with chemical fertilizers produced higher P fractions namely, S-P, Fe-P, and Ca-P, than alone use of compost and 100% RDF. Significant improvement had reported

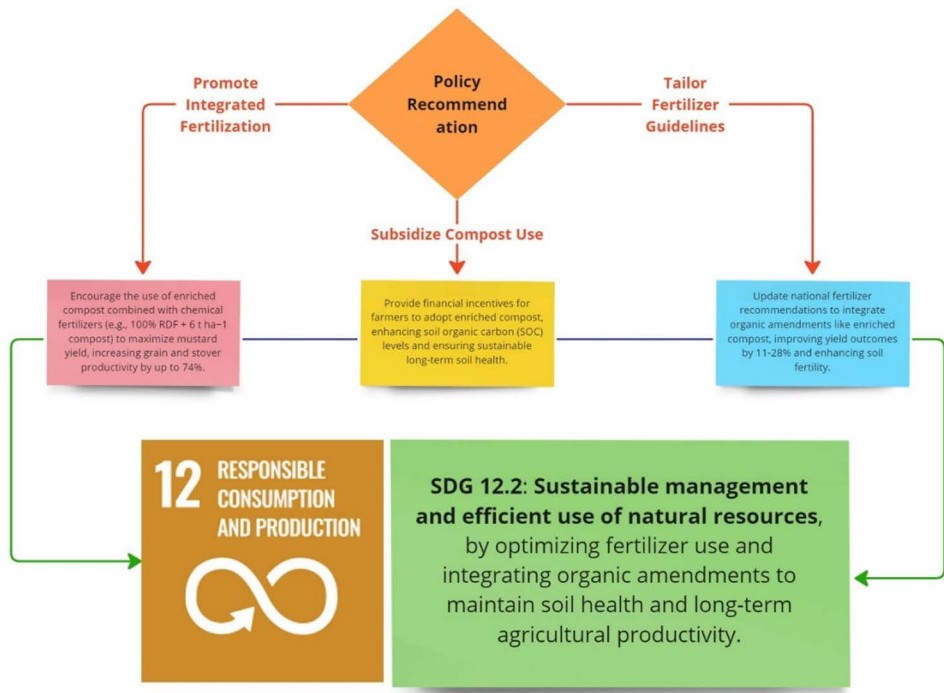

**Fig 11. Policy recommendations for sustainable mustard production.**

in organic-P (232 mg/ kg), MBP (5.83 mg/ kg), MBC (206 mg/ kg), alkaline phosphatase activity (159 g PNP/ g soil/24 h), and dehydrogenase activity (18 g TPF/ g soil/ h) in soil treated with enriched compost at 6 t/ha + 100% RDF. Compared to 100% RDF, organically treated soil had a greater amount of accumulated SOC. Results showed that integration of organic and chemical fertilizers increased soil P status and increased mustard productivity. However, enriched compost does not completely replace the water-soluble P fertilizers but continuous use of compost offered opportunities favouring soil P accumulation and improved soil health in the long run, making it a potential substitute for costlier chemical P fertilizers.

## Acknowledgments

The authors are thankful to the ICAR-Directorate of Rapeseed-Mustard Research, Sewar, Bharatpur, Rajasthan, India for excellent technical assistance.

## Author contributions

**Conceptualization:** Murli Dhar Meena, Vidhya Nand Mishra, Anup Kumar.

**Data curation:** Murli Dhar Meena, Mohan Lal Lakhera, Mukesh Kumar Meena, Mukesh Prajapat, Vasudev Meena.

**Formal analysis:** Dilkhush Meena, Mohan Lal Dotaniya, Mukesh Kumar Meena, Sujith Kumar, Mukesh Prajapat.

**Methodology:** Murli Dhar Meena, Mohan Lal Dotaniya.

**Resources:** Pramod Kumar Rai.

**Software:** Lalit Krishan Meena, Avijit Ghosh, Vijay Singh Meena.

**Supervision:** Satendra Singh Sengar, Vidhya Nand Mishra, Ram Swaroop Jat, Pramod Kumar Rai.

**Validation:** Murli Dhar Meena, Avijit Ghosh.

**Writing – original draft:** Dilkhush Meena, Murli Dhar Meena.

**Writing – review & editing:** Satendra Singh Sengar, Anup Kumar, Mohan Lal Lakhera, Mohan Lal Dotaniya, Sujith Kumar, Bheeru Lal Meena, Ram Swaroop Jat, Lalit Krishan Meena, Hari Singh Meena, Avijit Ghosh, Vasudev Meena, Pramod Kumar Rai, Parvender Sheoran, Vijay Singh Meena.

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
