## [Decision Letter · Decision Letter 0]

10 Mar 2025

PONE-D-25-00296Changes in soil phosphorus dynamics amended with rock phosphate-enriched compost and chemical fertilizersPLOS ONE

Dear Dr. Meena,

Thank you for submitting your manuscript to PLOS ONE. After careful consideration, we feel that it has merit but does not fully meet PLOS ONE’s publication criteria as it currently stands. Therefore, we invite you to submit a revised version of the manuscript that addresses the points raised during the review process.

We look forward to receiving your revised manuscript.

Kind regards,

Babar Iqbal, PhD

Academic Editor

PLOS ONE

Journal Requirements:

“The work was funded partially by the Director of the ICAR-Directorate of the Rapeseed-Mustard Research Institute Sewar, Bharatpur, Rajasthan, India, for which the authors are grateful. For financial support during the tenure in the form of a fellowship from ICAR, the senior author is also extremely grateful.”

**Additional Editor Comments:**

Kindly incorporate all the comments and addressed them one by one.

Reviewers' comments:

Reviewer's Responses to Questions

**Comments to the Author**

1. Is the manuscript technically sound, and do the data support the conclusions?

Reviewer #1: Yes

Reviewer #2: Yes

2. Has the statistical analysis been performed appropriately and rigorously? 

Reviewer #1: Yes

Reviewer #2: Yes

3. Have the authors made all data underlying the findings in their manuscript fully available?

Reviewer #1: Yes

Reviewer #2: Yes

4. Is the manuscript presented in an intelligible fashion and written in standard English?

Reviewer #1: Yes

Reviewer #2: Yes

5. Review Comments to the Author

Reviewer #1: Dear Editor

Thanks for your invitation to review the interesting article. Before consideration to publish the article, authors should revise it as per the list of comments:

1. Title should be changed to 'Changes in soil phosphorus dynamics as influenced by incorporation of rock phosphate-enriched compost and inorganic fertilizers'

2. Abstract: Authors used many abbreviations. I suggest to make a list of Abbreviations at the end of Abstract or as per the style of the journal. Mg ha-1 may be write as Mg/ha in whole manuscript as well as in all Figures and Tables

3. Recent findings/citations should be incorporated in the section. Also authors should mention actual objectives at the end of the section.

4. M & M: methods used in study should be written clearly for reader

5. Results: I am happy with the findings of the study, just authors should cite each Table and Figure in appropriate place

6. Similar to Introduction, latest citations should be added

Finally, I recommended it to publish, but after a major revision.

Reviewer

Reviewer #2: This study explored the impact of phosphorus management strategies on mustard production. It found that the combination of 100% recommended fertilizer dose (RDF) with enriched compost significantly improved phosphorus availability and soil microbial activity, leading to a 16.7% higher mustard grain yield compared to control. The results suggest that this combination enhances P accumulation and soil-plant P transformations, benefiting yields in degraded soils. Enriched compost at 6 t ha-1 also improved microbial biomass and enzymatic activity, supporting sustainable agricultural practices. The work is a meaningful contribution to agricultural modeling and is suitable for publication with minor revisions. (File attached)

6. PLOS authors have the option to publish the peer review history of their article (what does this mean? ). If published, this will include your full peer review and any attached files.

**Do you want your identity to be public for this peer review?** For information about this choice, including consent withdrawal, please see our Privacy Policy .

Reviewer #1: No

Reviewer #2: No

---

## [Author Response · Author response to Decision Letter 1]

26 Mar 2025

Response to Reviewers

RESPONSES TO THE COMMENTS/ SUGGESTIONS MADE BY YOU AND REVIEWER (S) ON THE MANUSCRIPT FOR PUBLICATION IN PLOS ONE

Ref No: PONE-D-25-00296

Title: Changes in soil phosphorus dynamics amended with rock phosphate-enriched compost and chemical fertilizers

Sir, first of all, I would like to thank you and the reviewers for your critical but encouraging comments/suggestions on our paper entitled “Changes in soil phosphorus dynamics amended with rock phosphate-enriched compost and chemical fertilizers” submitted by M.D. Meena and others for consideration in PLOS ONE after addressing all comments.

Once again, I thank you and the reviewer (s) for your comments/suggestions on our paper.

As suggested by you, I have incorporated responses of each comment in track change mode in the revised manuscript.

Point-wise responses to the comments/suggestions made by reviewers and the editor are as follow:

Editorial Comments

The manuscript was revised as per the PLOS ONE guideline

“The work was funded partially by the Director of the ICAR-Directorate of the Rapeseed-Mustard Research Institute Sewar, Bharatpur, Rajasthan, India, for which the authors are grateful. For financial support during the tenure in the form of a fellowship from ICAR, the senior author is also extremely grateful.”

The acknowledgments section in the manuscript has been modified as The authors are thankful to the ICAR-Directorate of Rapeseed-Mustard Research, Sewar, Bharatpur, Rajasthan, India for excellent technical assistance in the revised manuscript.

3. We note that you have indicated that there are restrictions to data sharing for this study. PLOS only allows data to be available upon request if there are legal or ethical restrictions on sharing data publicly. For more information on unacceptable data access restrictions

We have mentioned that all the data are available within the manuscript in the data availability section

As suggested by you and the reviewer (s) we have incorporated in the revised manuscript as suggested by you.

We have incorporated captions for figures and tables as per the journal guideline

Reviwer#1

1 The title is very suitable for the study, no need to change the title.

Thank you very much comment, we have not modified the title in the revised manuscript as suggested by you.

2. The abstract of the study is sufficient.

No changes were incorporated in the revised MS except abbreviations.

3. Line 6, the abbreviation should be elaborated when using first time.

We have elaborated the abbreviation the first time and the subsequent short form in the revised manuscript

4. Line 8, change the “we” and check the grammar.

We have modified the sentence as suggested by you in the revised MS

5. Please arrange the Keywords in alphabetical order.

Keywords are arranged as suggested by you.

6.The introduction is inappropriate. The first paragraph is overly long, while the others are too short. Please review and reorganize the introduction section. Adjust the language to ensure better clarity and coherence. Also introduce Mustard juncea in few lines.

The introduction has been modified and reorganized some sentences and also incorporated the importance of mustard (Brassica juncea L.) crop in the revised manuscript.

7. Line number 108, Reference is not according to journal format.

The reference has been modified in the revised manuscript as per the journal guideline

8. The study shows that combining enriched compost (6 t ha⁻¹) with 100% RDF boosts Olsen-P, total P, inorganic P, and microbial activity compared to the control. However, enriched compost alone cannot fully replace water-soluble P fertilizers. Could the authors explain the limitations of enriched compost in replacing chemical P fertilizers? Also, what are the long-term effects of using enriched compost for P availability in soils with low P content or high P fixation?

Enriched compost is slow-release in nature in terms of nutrients and alone it cannot supply the nutrient demand as required by the crop in the first year as compared to chemical fertilizers because it has more water-soluble phosphorus which is easily taken up by plants. However, in the subsequent years, the enriched compost increased nutrient-supplying potential to plants for long-term perspective because of the faster mineralization rate.

9. There is a problem with using abbreviations. The full term should be mentioned first with the abbreviation between paresis then the abbreviations should be exclusively used throughout the manuscript. Please revise the whole paper’s abbreviations carefully and correct them accordingly.

We have thoroughly revised the abbreviations in the first time used full form and subsequently abbreviated in the whole MS

10. I reviewed the layout of the article: I would suggest the tables and figures should appear where they are mentioned in the text the first time (in Results) and not several pages later.

The tables and figures are incorporated into the text in the revised manuscript as suggested by you.

11. Please use the same and uniform text size of heading and footnotes for all Tables and Figures throughout the paper.

We have used the uniform font size of headings and footnote in all tables and figures in the revised manuscript

12. Please avoid symbols at the beginning of the sentences.

We have removed the symbols at the beginning of the sentences in the revised manuscript.

13. Revise the conclusion and summarize it according to the key findings of your study.

The conclusions have been summarized as per the salient findings of the study

14. Please arrange all the references according to the Journal format.

All the references are arranged according the journal format

Reviwer#2

1. Title should be changed to 'Changes in soil phosphorus dynamics as influenced by incorporation of rock phosphate-enriched compost and inorganic fertilizers'

The title of the manuscript seems okay

2. Abstract: Authors used many abbreviations. I suggest to make a list of Abbreviations at the end of Abstract or as per the style of the journal. Mg ha-1 may be write as Mg/ha in whole manuscript as well as in all Figures and Tables

The list of abbreviations has been incorporated after the abstract and units have also modified in revised manuscript as well as all figures/ tables as suggested by you.

3. Recent findings/citations should be incorporated in the section. Also authors should mention actual objectives at the end of the section.

The latest citations incorporated in revised manuscript

4. M & M: methods used in the study should be written clearly for reader

Methods and materials have been modified as suggested by you.

5. Results: I am happy with the findings of the study, just authors should cite each Table and Figure in appropriate place

The tables and figures cited in text in the revised manuscript.

6. Similar to Introduction, latest citations should be added

In the introduction section latest references were incorporated in the revised manuscript

Once again, I thank you very much sir for your valuable comments/suggestions for publication in PLOS ONE.

With kind regards

Yours sincerely,

Dr. M.D. Meena

Senior Scientist

ICAR-Directorate of Rapeseed-Mustard Research, Sewar,

Bharatpur-321303, India

Mobile: +91-9896646387

(E-mail: murliiari@gmail.com).

---

## [Decision Letter · Decision Letter 1]

30 Apr 2025

Changes in soil phosphorus dynamics amended with rock phosphate-enriched compost and chemical fertilizers

PONE-D-25-00296R1

Dear Dr. Murli ,

We’re pleased to inform you that your manuscript has been judged scientifically suitable for publication and will be formally accepted for publication once it meets all outstanding technical requirements.

Kind regards,

Babar Iqbal, PhD

Academic Editor

PLOS ONE

Additional Editor Comments (optional):

I appreciate the authors for revising the manuscript. Thus, I recommend accepting it in its current form.

Reviewers' comments:

Reviewer's Responses to Questions

**Comments to the Author**

1. If the authors have adequately addressed your comments raised in a previous round of review and you feel that this manuscript is now acceptable for publication, you may indicate that here to bypass the “Comments to the Author” section, enter your conflict of interest statement in the “Confidential to Editor” section, and submit your "Accept" recommendation.

Reviewer #2: All comments have been addressed

Reviewer #3: All comments have been addressed

2. Is the manuscript technically sound, and do the data support the conclusions?

Reviewer #2: Yes

Reviewer #3: Yes

3. Has the statistical analysis been performed appropriately and rigorously? 

Reviewer #2: Yes

Reviewer #3: Yes

4. Have the authors made all data underlying the findings in their manuscript fully available?

Reviewer #2: Yes

Reviewer #3: Yes

5. Is the manuscript presented in an intelligible fashion and written in standard English?

Reviewer #2: Yes

Reviewer #3: Yes

6. Review Comments to the Author

Reviewer #2: (No Response)

Reviewer #3: (No Response)

7. PLOS authors have the option to publish the peer review history of their article (what does this mean? ). If published, this will include your full peer review and any attached files.

**Do you want your identity to be public for this peer review?** For information about this choice, including consent withdrawal, please see our Privacy Policy .

Reviewer #2: No

Reviewer #3: No

---

## [Editor Report · Acceptance letter]

PONE-D-25-00296R1

PLOS ONE

Dear Dr. Meena,

I'm pleased to inform you that your manuscript has been deemed suitable for publication in PLOS ONE. Congratulations! Your manuscript is now being handed over to our production team.

Kind regards,

on behalf of

Dr. Babar Iqbal

Academic Editor

PLOS ONE